# Physical Activity Level and Self-Esteem in Middle-Aged Women

**DOI:** 10.3390/ijerph18147293

**Published:** 2021-07-08

**Authors:** Magdalena Dąbrowska-Galas, Jolanta Dąbrowska

**Affiliations:** 1Department of Kinesitherapy and Special Methods, School of Health Sciences in Katowice, Medical University of Silesia, 40-752 Katowice, Poland; 2School of Health Sciences in Katowice, Medical University of Silesia, 40-752 Katowice, Poland; 1dabrowskajolanta@gmail.com

**Keywords:** self-esteem, IPAQ, MRS, BDI, menopause

## Abstract

*Background*: Physical activity (PA) is a behavioral modality that may help decrease negative symptoms of menopause and enhance some positive aspects of mental health, including self-esteem. Reduced self-esteem may put menopausal women at higher risk of negative outcomes of menopause and result in a more unpleasant and stressful menopausal experience. The objective of this study was to examine the role of physical activity level on self-esteem in middle-aged women. *Methods*: Women aged 45–60 from Poland took part in this study. The Rosenberg Self-Esteem Scale, the International Physical Activity Questionnaire, the Menopause Rating Scale and Beck Depression Inventory were used in this study. *Results*: Among the 111 women, the mean age was 51.7 ± 4.7. The most severe symptoms among studied women concerned sexual problems (1.71 ± 1.5), irritability (1.58 ± 1.37) and joint and muscular discomfort (1.56 ± 1.55). Women with higher total PA level had better self-esteem (*p* = 0.001). *Conclusions:* The results of this study showed that physical activity levels can be associated with self-esteem. Most middle-aged women reported high physical activity levels. These results have clinical implications for the inclusion of PA in the lives of middle-aged women to improve self-esteem and mental health.

## 1. Introduction

An aspect of a woman’s health in the middle-age period has attracted the attention of many researchers. Women spend half of their life in the middle age and afterwards, and during this time they experience physical and mental changes, as well as undergoing social role changes [1].

Menopausal transition has been related to a number of health impairments and variability of symptoms women experience as a result of hormone changes [2]. These, in turn, can affect self-esteem and life satisfaction [3]. The media create an ideal image of women, and all exceptions related to the inconveniences associated with menopausal symptoms and mental or physical disorders are sometimes stigmatized [4]. The results of Hunter and Rental (2007) showed that women with a higher severity of vasomotor symptoms and experiencing stress, report lower self-esteem [5]. Bloch (2002) showed that the severity of psychological symptoms decreased the self-esteem of menopausal women [6]. Włodarczyk et al. (2017) showed that women with high intensity of psychological, vasomotor and somatic menopausal symptoms had lower self-esteem [7]. Lower self-esteem is identified as one of the most common negative symptoms during this time [8,9].

Self-esteem is an essential component of well-being, an important factor of emotional and social adjustment, and is defined as an individual belief that one is a “good enough” and valuable person [10,11]. People with lower levels of self-esteem appear to be more susceptible to negative effects such as anxiety, lack of satisfaction, lower body esteem and depression [12,13]. Reduced self-esteem may put menopausal women at higher risk for negative outcomes of menopause and result in a more unpleasant and stressful menopausal experience [14]. Nosek et al. (2010) showed that high self-esteem helps with coping with the transition through the menopause [15].

Physical activity (PA) is one behavioral modality that may help decrease negative symptoms of menopause and enhance some positive aspects of mental health, including self-esteem [16,17,18]. Recent studies of PA and self-esteem have suggested that physical self-esteem was a part of global self-esteem and was related to body attractiveness, physical condition and strength; moreover, some studies have suggested that physical appearance or body esteem may even be synonymous with global self-esteem [19].

Additionally, menopausal women often have a negative body image, which in turn correlates with low self-esteem [20]. The menopausal transition is also related to weight gain and increases in central adiposity [13,21]. Previous studies suggest that among middle-aged women there is a relationship between BMI and satisfaction with body esteem, thus, body changes during menopause may place women at increased risk for diminished self-esteem [22,23]. Most studies focused on the relationship between weight changes and self-esteem, and showed that higher self-esteem was associated with weight loss or lower BMI [24,25]. Earlier research demonstrated a relationship between body image and global self-esteem, reporting that PA reduces BMI, improves body attractiveness and, consequently, self-esteem [23].

A few studies have examined the relationship between specific PA activities such as yoga, walking, gardening, jogging or cycling on self-esteem in women [26]. However, surprisingly, no research has been conducted examining the effects of physical activity level on self-esteem in middle-aged women. Therefore, the objective of this study was to examine the effect of physical activity level on self-esteem in middle-aged women.

## 2. Materials and Methods

### 2.1. Study Group

This was a cross-sectional study. Women aged 45–60 who visited healthcare center in Silesia in Poland, and agreed to participate in the study, were selected. Participation in the study was voluntary and anonymous. Verbal informed consent was obtained from all participants. The inclusion criteria for the research were: aged 45–60 years, consent to participate in the research, no serious illness. The exclusion criteria were contraindications to physical activity, impaired physical mobility, major depression, incomplete completed questionnaire. In total, 130 women were invited to participate in the study. Nineteen women with missing data in the questionnaire were excluded from the analysis. The study protocol was reviewed and approved by the Bioethical Committee of the Medical University of Silesia in Katowice (PCN/022/KB1/147/I/19/20).

### 2.2. Questionnaire

The research tool was a survey consisting of five parts (one self-designed survey and four validated, international questionnaires: the Rosenberg Self-Esteem Scale (RSES), the International Physical Activity Questionnaire short form (IPAQ-SF), the Menopause Rating Scale (MRS) and the Beck Depression Inventory-Second Edition (BDI-II). The first part included self-designed questions about sociometric and basic gynecologic data. 

#### 2.2.1. The Rosenberg Self-Esteem Scale

This questionnaire was used to evaluate self-esteem. This scale comprises 10 statements, with four choices of answers on the Likert point scale (1 = completely agree; 4 = completely disagree). The scale has two factors. The first one comprises six items related to positive self-esteem, and the second factor, four items that refer to negative self-esteem. Total score ranges from 10 to 40, the higher the score on the scale, the higher the individual level of self-esteem [27].

#### 2.2.2. The International Physical Activity Questionnaire

This tool was used to assess the level of physical activity. IPAQ is a self-report questionnaire evaluating physical activity level. It is intended for adults aged 15–69 years. It consists of questions about the previous 7 days and asks about the frequency and duration of low, moderate and high intensity PA lasting at least 10 min [28]. PA level was expressed as Metabolic Equivalent of Work (MET)-min (where the metabolic equivalent of 1MET is defined as the amount of oxygen consumed when sitting at rest and is 3.5 mL O_2_ per kg body weight/min) per week. According to the IPAQ scoring protocol, women were classified into groups with low, moderate or high levels of physical activity, and weekly energy expenditure for each PA level was calculated by multiplying the MET by number of days during the week. In accordance with IPAQ methodology, vigorous PA was assigned to 8 MET, moderate PA to 4 MET and walking to 3 MET [29].

#### 2.2.3. The Menopause Rating Scale

This validated and standardized scale was used to assess the severity of menopausal symptoms. The scale ensures documented credibility, sensitivity, reliability and duplication of results. MRS is a self-administered tool consisting of 11 descriptions of climacteric symptoms. All items are divided into 3 domains: psychological (depressive mood, physical and mental exhaustion, irritability and anxiety), somato-vegetative (hot flushes, heart discomfort, muscle and joint problems and sleeping problems), urogenital (sexual problems, dryness of vagina and bladder problems).

Total score is the sum in each of the domains and ranges from 0 (asymptomatic) to 44 (highest degree of complaints) [30,31].

#### 2.2.4. The Beck Depression Inventory-Second Edition

This inventory was used to evaluate the presence and severity of depressive symptoms [32]. This questionnaire is a 21-item self-reporting screening tool, with each item rated on a 4-point scale from 0 (not present) to 3 (severe). The total score ranges from 0 to 63, with higher values indicating more severe depressive symptoms. The BDI-II score from 0 to 13 was interpreted as no depressive symptoms, 14 to 19 as mild depression, 20–28 as moderate depression and 29–63 as severe depression [32].

### 2.3. Statistical Analysis

Statistical analysis was performed using the Statistica 10 (Statistica v10, StatSoft, Krakow, Poland). For measurable variables, such values as arithmetic means, median, standard deviations were calculated. For qualitative variables, percentage was calculated. The Shapiro–Wilk test was used to determine normality of data distribution. The analysis of variance (ANOVA) with post hoc Tuckey test was used. The level of α = 0.05 was assumed as statistically significant.

## 3. Results

Among 111 women, the mean age was 51.7 ± 4.7. The mean BMI value of women was 26.12. The majority of women were living in a city of up to 100,000 citizens (47.75%), did not smoke (75.68%), were higher educated (45.05%), married or cohabitating (80.18%) and had irregular periods (46.85%). According to IPAQ, 41.44% of women had a high PA level, 31.53% of participants had an average PA level, while 27.03% presented a low PA level. The majority of women had no depressive symptoms (82.88%). The results showed that 15.32% of women reported very high self-esteem. Approximately one-third of participants had high (29.73%) or average (24.32%) self-esteem and one-fifth had very low self-esteem (Table 1).

The results of the MRS showed that the most severe symptoms among studied women concerned sexual problems (1.71 ± 1.5), irritability (1.58 ± 1.37) and joint and muscular discomfort (1.56 ± 1.55). Sleep problems and physical and mental exhaustion were evaluated at the similar mean value (1.46 and 1.48, accordingly). The mean value of depressive mood was 1.11 (Table 2). According to MRS domains (urogenital, somato-vegetative and psychological), the highest degree of complaints were observed in the somato-vegetative domain (5.6 ± 3.69), while the lowest degree of complains were reported in the urogenital domain (3.82 ± 3.4). The psychological domain had a mean value of 5.6 ± 3.69 (Table 2).

A significant relationship was observed between self-esteem and the physical activity level in specific PA levels and in total (Table 3). Women with higher total PA level had better self-esteem (*p* = 0.001). Significant between-group differences in self-esteem and PA level were observed between very low and high self-esteem, as well as very low and very high self-esteem. According to the moderate and high PA level in relation to self-esteem, the higher the self-esteem the higher mean value of MET-min/week was observed. Specific intergroups relations are presented in Table 3. In terms of walking, women with high self-esteem had a slightly higher walking MET-value than the participants with very high self-esteem; however, no significant relation between these two groups was observed.

## 4. Discussion

The North American Menopause Society (NAMS) recommends changing lifestyle in terms of increasing physical activity as the first line of defense against the adverse consequences of menopause [33]. Physical activity has a direct impact on many adverse consequences of menopause and has been shown to improve self-esteem [16,17,18]. Women with low self-esteem report more severe menopausal symptoms and worse quality of life, while healthy self-esteem is an important component of mental health [14,19].

In the present study, we evaluated the physical activity level and self-esteem in middle-aged women. We were interested in examining whether PA level influenced self-esteem. We classified physical activity level into low, moderate and high, according to IPAQ. The results were surprising because 41.44% of women reported a high PA level. Our results are consistent with the previous study from Poland [18].

However, earlier studies have shown a reduction in daily energy expenditure and a shift toward a more sedentary lifestyle during the menopausal transition [34,35]. A cross-sectional study from Finland showed that the majority of women aged 47–55 reported a moderate PA level (62.32%) [36]. A study from Poland showed similar results [15]. Another large cross-sectional study from Japan showed that 57.84% of perimenopausal women (aged 44–56) had a low PA level [37]. Our study included women in the 45–65 age group, thus the difference in PA level may result from age. It may indicate that postmenopausal women are more physically active.

Different studies have shown a significant decline in body muscle power and handgrip strength in postmenopausal women when compared to premenopausal women [38,39]. However, Bondarev et al. (2018) showed no difference in a 6-min walking test distance in three menopause status groups [36]. This may indicate that menopausal status does not affect the cardiovascular and respiratory system. In another study carried out on middle-aged women, no decline in maximal oxygen consumption during the menopausal transition was observed [40].

To the best of our knowledge, no previous studies have concentrated on the level of PA according to self-esteem in middle-aged women. Our results revealed significant differences between PA level and self-esteem in women. For moderate and high PA levels, higher self-esteem in women was reported together with a higher mean value of MET-min/week. It is interesting that the mean value of MET-min/week in walking was highest when compared with the moderate or high PA level. Even women with very low self-esteem reported 353.93 MET-min/week walking compared to 144 MET-min/week in the high PA level group or 261 MET-min/week in the moderate PA level group. A similar relation was observed in all self-esteem levels. Walking MET-min/week showed the highest value. This may indicate that walking is a highly prevalent and preferred form of PA among middle-aged women.

Elavsky et al. (2007) enrolled middle-aged women in a 4-month randomized trial with yoga exercises and walking. The analysis of the follow-up study showed a small improvement in self-esteem in the walking and yoga groups compared to a control group [25]. These findings suggest that non-aerobic type exercise such as yoga or walking at low intensities are effective for the enhancement of self-esteem.

Self-esteem in middle-aged women was directly analyzed in a few previous studies; specifically, the relationship between body esteem and self-esteem in middle-aged women was evaluated, and the results were mixed. Some studies have reported that body esteem was not related to global self-esteem and that BMI or body image did not play a significant role [27,41]. Contrary to this, different studies showed that more active middle-aged women reported significantly higher body esteem and global self-esteem [19,42]. A longitudinal study evaluating the exercise and self-esteem mode in middle-aged women also showed that women with lower BMI reported a higher level of physical self-worth, which was related to global self-esteem [43].

Our study has several limitations. Firstly, it was a cross-sectional study, thus, a proper randomization was not achieved and there is a possibility of a selection bias. Such a study also gives restrictions on drawing causal effects. Secondly, the results should be interpreted with caution because the study sample is too small to generalize the obtained results. Thirdly, self-reported questionnaires were used; however, these are the usual methods in the literature. These findings may have been influenced by the common methods bias. In the future, a longitudinal study conducted on a representative sample will provide a detailed analysis showing the impact of different variables on self-esteem in women.

## 5. Conclusions

The results of this study showed that physical activity levels are associated with self-esteem. Middle-aged women with higher physical activity levels had better self-esteem. Most middle-aged women reported high physical activity levels.

These results have clinical implications for the inclusion of PA in the lives of middle-aged women to improve self-esteem and mental health. Middle-aged women with low self-esteem may benefit from increasing their PA level, for example, by making walking a routine. A further study in a larger, representative sample of middle-aged women is warranted.

## Figures and Tables

**Table 1 ijerph-18-07293-t001:** Characteristics of all participants.

Variables		Study Group (*n* = 111)			
		Mean	Min	Max	SD
Age (years)		51.7	45	60	4.7
Weight (kg)		70.37	49	100	11.48
Hight (cm)		164.05	150	176	5.72
BMI (kg/m^2^)		26.12	16.96	34.6	3.86
		*n*		%	
Place of residence	Village	15		13.51	
	A city of up to 100,000	53		47.75	
	A city above 100,000	43		38.74	
Smoking	No	84		75.68	
	Yes	27		24.32	
Education	Primary	20		18.02	
	Secondary	41		36.94	
	Higher	50		45.05	
Marital	Married/	89		80.18	
status	Cohabitating				
	Single	7		6.31	
	Divorced	11		9.91	
	Widow	4		3.6	
Physical	Low	30		27.03	
activity level	Moderate	35		31.53	
(IPAQ)	Higher	46		41.44	
BDI	No	92		82.88	
	Mild	11		9.91	
	Moderate	6		5.41	
	Severe	2		1.80	
SES	Very low	20		18.02	
	Low	14		12.61	
	Average	27		24.32	
	High	33		29.73	
	Very high	17		15.32	

BDI—Beck Depression Inventory; SES—Self-esteem Scale; IPAQ—International Physical Activity Questionnaire.

**Table 2 ijerph-18-07293-t002:** The characteristics of menopausal symptoms (MRS scale) in the participants.

Symptoms (MRS Scale)	Mean	±SD
Hot flushed, sweating	1.30	1.28
Heart discomfort	1.29	1.36
Sleep problems	1.46	1.56
Depressive mood	1.11	1.23
Irritability	1.58	1.37
Anxiety	0.87	1.17
Physical and mental exhaustion	1.48	1.32
Sexual problems	1.71	1.50
Bladder problems	0.72	1.16
Dryness of vagina	1.39	1.50
Joint and muscular discomfort	1.56	1.55
Urogenital domain	3.82	3.40
Psychological domain	5.04	3.86
Somato-vegetative domain	5.60	3.69
Total	14.46	8.99

MRS—Menopause Rating Scale.

**Table 3 ijerph-18-07293-t003:** Relationship between physical activity level (IPAQ) and self-esteem (SES).

	SES										
	Very Low		Low		Average		High		Very High		*p*
IPAQ	Mean	SD	Mean	SD	Mean	SD	Mean	SD	Mean	SD	
total MET-min/week	758.9 * ᶷ	1223.13	1354.64	2090.48	1881.26	2304.79	4508.03 *	6333.23	5632.47 ᶷ	4969.70	0.001
high PA level MET-min/week	144.00 ᶷ	378.67	214.29 †	456.77	400.00 ^Ƹ^	1138.91	789.09	1614.85	1927.06 † ᶷ ^Ƹ^	2470.76	0.002
moderate PA level MET-min/week	261.00 * ᶷ	767.54	374.29 ^	368.15	464.44 ~ ^Ƹ^	593.87	1336.97 * ~	2268.52	1717.65 ᶷ † ^Ƹ^	2079.54	0.009
walking PA level MET-min/week	353.93 *	470.59	766.07	1842.79	1016.89 ^Ƹ^	1447.99	2382.00 *	3306.89	1987.76 ^Ƹ^	2565.75	0.014

* *p* < 0.05 very low versus high; ᶷ *p* < 0.05 very low versus very high; † *p* < 0.05 low versus very high; ^Ƹ^ *p* < 0.05 average versus very high; ^ *p* < 0.05 low versus high; ~ *p* < 0.05 average versus high; IPAQ—International Physical Activity Questionnaire. SES—Self-esteem Scale, Rosenberg.

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
