# Peer review of "Physical Activity Level and Self-Esteem in Middle-Aged Women"

_ijerph, 2021, doi:10.3390/ijerph18147293_

Round 1
Reviewer 1 Report
Overall excellent research and article. A few comments
In the abstract line 18 PA has not been written out yet. Perhaps this is OK in an abstract?
Introduction lines 25-60 excellent use of references.
Lines 64-67 inclusion and exclusion criteria. No serious illness is stated but impaired physical mobility should be considered as an exclusion, also major depression. Also thoughts on excluding those on hormone replacement, which negates the symptoms of menopause.
Good selection of assessment instruments.
Results – Any consideration of correlation between depression and self-esteem?
Would have liked to see a section specific to the physical activity. What types did the participants take part in? Such as x number walked, x biked etc. And length of time. Did they do the activity alone or with others in a group.
Overall most of my comments could be considered in follow up research as this is a good starting point for Physical exercise and self-esteem.
Unless authors have any information on the comments I addressed the only thing needed is addressing line 18.
Great job. Accept as is except for correcting line 18.but
Author Response
Overall excellent research and article. A few comments
In the abstract line 18 PA has not been written out yet. Perhaps this is OK in an abstract?
Response: Thank you very much for your kind words about this paper. Thank you for pointing this out. We have added in abstract the abbreviation in the first line, thank you for this suggestion (line 9).
Introduction lines 25-60 excellent use of references.
Response: Thank you.
Lines 64-67 inclusion and exclusion criteria. No serious illness is stated but impaired physical mobility should be considered as an exclusion, also major depression. Also thoughts on excluding those on hormone replacement, which negates the symptoms of menopause.
Good selection of assessment instruments.
Response: Thank you for your appreciation of instruments used in this study and for your suggestions about exclusion criteria. We have added impaired physical mobility and major depression. None of the participants had major depression or impaired physical mobility. We will exclude women taking HT in future, thank you for this comment.
Results – Any consideration of correlation between depression and self-esteem?
Would have liked to see a section specific to the physical activity. What types did the participants take part in? Such as x number walked, x biked etc. And length of time. Did they do the activity alone or with others in a group.
Response: Thank you for this suggestion. We are also very interested in specific type of exercise that would influence self-esteem. However, IPAQ does not give a possibility to report specific type of exercise. According to IPAQ we ask about time spent on high, moderate PA or walking. In future study we will add such questions to the survey to find out what kind of exercises are commonly chosen by women and how it influences self-esteem. We have done a correlation between depression and self-esteem, however, it was insignificant. Maybe because 83% had no depression symptoms. Future studies are warranted
Overall most of my comments could be considered in follow up research as this is a good starting point for Physical exercise and self-esteem.
Unless authors have any information on the comments I addressed the only thing needed is addressing line 18.
Great job. Accept as is except for correcting line 18.
Response: It is very pleasant to hear such words about our manuscript. The abbreviation (PA) has been added in the first line. All other suggestions will be considered in future research to improve the study.
Reviewer 2 Report
We now physical activity level had better self-esteem in all age
The authors should have an average age control group, perhaps between 20 and 40 years and verify whether the levels of self-esteem increase were different in both groups and whether this difference was significantly different.
Wanting to work only with middle-aged populations and decrease negative symptoms in menopause, perhaps it would be more productive for science, accurately identify which exercises had the greatest emotional and also physiological effect (lower blood pressure, lower cholesterol). With these exercises make a table of exercises also identifying the impact of exercises, minimum impact, medium impact, high impact.
Author Response
We now physical activity level had better self-esteem in all age.
Response: Many thanks for these comments and time and effort you have put into this manuscript. We have analyzed literature and found that major results show positive impact of physical activity on self-esteem, however, some publications showed no correlation between physical activity and self-esteem in females. We kindly consider all your suggestions in future research because more studies are needed to clarify this issue.
The authors should have an average age control group, perhaps between 20 and 40 years and verify whether the levels of self-esteem increase were different in both groups and whether this difference was significantly different.
Response: Thank you for this suggestion. This topic is very interesting for us and we agree that such data would show interesting results. We will continue this research. In the follow-up study we will provide a control group as recommended.
Wanting to work only with middle-aged populations and decrease negative symptoms in menopause, perhaps it would be more productive for science, accurately identify which exercises had the greatest emotional and also physiological effect (lower blood pressure, lower cholesterol). With these exercises make a table of exercises also identifying the impact of exercises, minimum impact, medium impact, high impact.
Response: Thank you for pointing this out. It is extremely vulnerable suggestion and we agree that it may show interesting results. However, in our study we did not have such data as blood pressure, cholesterol level or type of exercises. According to IPAQ scoring protocol we divided participants into 3 groups: with low, moderate and high PA level (Table 3). In future study we will include all suggestions and add questions about specific type of exercises and more questions about physiological aspect. Such database with specific exercises will allow to provide results with specific impact on self-esteem. If possible, we will consult with you in the future study with a great pleasure in order to prepare a perfect questionnaire including all important variables.
Reviewer 3 Report
ijerph-1261483- peer-review-v1
Is the article within the scope of the journal? The paper is within the scope of the journal. It is an interesting topic to examine physical activity level and self-esteem in middle aged women.
The importance of subject matter and purpose?
The article addresses and explores an interesting topic. However, the study seems to be based on a special group of women with high education, few smokers and a low BMI average. The latter is relatively unusual for being in the middle-age. There also seems to be underweight or something else in the group.
- Abstract
The abstract describes the different parts and the process in an acceptable way. Perhaps you should add the word “may” to the sentence “The results of this study showed that physical activity levels may be associated with self-esteem.”
Introduction section
- Se below
Adherence, to ethical standards?
- The ethical considerations are not clearly mentioned.
Contribution of new knowledge?
· The article contributes to the knowledge but needs some clarifications.
My main concern and query with the article is the following:
- Introduction: The text could be more clearly written e. g. in more detail describing the concept of self-esteem in relation to how it affects menopause.
- Aim: There is an ambiguity in what the research purpose is. the text presents both the effect of physical activityand the role of the level of physical activity: “no research has been conducted examining the effects of physical activity level on self-esteem in middle-aged women. Therefore, the objective of this study was to examine the role of physical activity level on self-esteem in middle-aged women” (line 57-60).
- Materials and Methods: This section is not sufficiently developed. In the text there are mixes of information about the instrument itself and how it is used with how the actual collection of data took place. One proposal to make this text clearer is to introduce a special heading under which the instruments themselves are described and used and then have unother heading with how the actual collection took place for this article.
- Result: The presentation of the result could be more clear, for example highlight what is particularly interesting in relation to the research question. It may also be helpful to clarify why the points sexual problems, irritability, and muscle problems are highlighted when other factors such as urogenitis, psychological, and somato-vegetative domains appear to be major problems.
- Discussion. This is the best part of the article. In addition to the limitations mentioned at the end of the discussion, it could be
In general
- How do this affects the result?
- Sometimes the sentences in the text are short and undeveloped.
Add year to e.g. Elavsky et
Author Response
Is the article within the scope of the journal? The paper is within the scope of the journal. It is an interesting topic to examine physical activity level and self-esteem in middle aged women.
The importance of subject matter and purpose?
The article addresses and explores an interesting topic. However, the study seems to be based on a special group of women with high education, few smokers and a low BMI average. The latter is relatively unusual for being in the middle-age. There also seems to be underweight or something else in the group.
- Abstract
The abstract describes the different parts and the process in an acceptable way. Perhaps you should add the word “may” to the sentence “The results of this study showed that physical activity levels may be associated with self-esteem.”
Introduction section
- Se below
Adherence, to ethical standards?
- The ethical considerations are not clearly mentioned.
Contribution of new knowledge?
- The article contributes to the knowledge but needs some clarifications.
Response: Thank you for your time and suggestions, we strongly appreciate your comments. In accordance with your suggestions we have revised a sentence in the Abstract into “The results of this study showed that physical activity levels may be associated with self-esteem” – as recommended.
We have divided “Material and methods’ section into: Study Group, Questionnaire, Statistical Analysis. In the “Study Group” section a sentence: “Verbal informed consent was obtained from all participants” has been added.
My main concern and query with the article is the following:
- Introduction:The text could be more clearly written e. g. in more detail describing the concept of self-esteem in relation to how it affects menopause.
Response: We thank the review for suggestion and we agree. In a revised version of the manuscript we have added necessary information to introduction and hope it is more clear now.
- Aim:There is an ambiguity in what the research purpose is. the text presents both the effect of physical activity and the role of the level of physical activity: “no research has been conducted examining the effects of physical activity level on self-esteem in middle-aged women. Therefore, the objective of this study was to examine the role of physical activity level on self-esteem in middle-aged women” (line 57-60).
Response: Thank you for pointing this out. We have changed the statement (aim) into: “Therefore, the objective of this study was to examine the effect of physical activity level on self-esteem in middle-aged women”
- Materials and Methods:This section is not sufficiently developed. In the text there are mixes of information about the instrument itself and how it is used with how the actual collection of data took place. One proposal to make this text clearer is to introduce a special heading under which the instruments themselves are described and used and then have unother heading with how the actual collection took place for this article.
Response: We thank the reviewer for this viewpoint. We have divided “Material and methods’ section into: Study Group, Questionnaire, Statistical Analysis. In section “Questionnaire” there are heading will all questionnaires as recommended. All questionnaires used in the study are clearly explained. In this study data were collected according to instrument’s methods.
- Result:The presentation of the result could be more clear, for example highlight what is particularly interesting in relation to the research question. It may also be helpful to clarify why the points sexual problems, irritability, and muscle problems are highlighted when other factors such as urogenitis, psychological, and somato-vegetative domains appear to be major problems.
Response: We thank the reviewer for this suggestion. We need to explain the we have highlighted e.g sexual problems or irritability because these domains had the highest mean value. We agree that major problem was not mentioned. As recommended, we have added other, remeining domains description.
- This is the best part of the article. In addition to the limitations mentioned at the end of the discussion, it could be
In general
- How do this affects the result?
- Sometimes the sentences in the text are short and undeveloped.
Add year to e.g. Elavsky et
Response: We thank the reviewer for the appreciation of this section. As recommended, we have added information about the influence of limitations into our results. Year has been added to publications used in Discussion. We have revised a the text and a few sentences have been changed, as recommended.
Round 2
Reviewer 2 Report
Nothing to add. The justifications presented are clear
Author Response
Reviewer 2:
Thank you very much for accepting our justification and for sharing your experience and opinion with us. It helped us improve our manuscript.
Reviewer 3 Report
There is still some thing that needs to be considered e.g. line 55 research by Ayers B et al. (2010). Why the initial here? Why is et al,. not italic?
Still some sentences in the text are short and undeveloped e.g. in line 36 … showed that women with a higher severity of vasomotor symptoms experiencing stress, report lower self-esteem [5]. Recommend a “and” between stress, report …
But otherwise I think the manuscript is better
Author Response
Reviewer 3:
1.There is still some thing that needs to be considered e.g. line 55 research by Ayers B et al. (2010). Why the initial here? Why is et al,. not italic?
Response: We read the ‘Instruction for authors” very carefully. We did not find information about writing et al in italic, neither in Instruction for authors, nor in published articles in IJERPH. We would not like to confuse the Editor. We have added year to this initial as it was recommended by another reviewer. However, according to your suggestion, maybe it is better to delete both initial and year. We have revised the sentence:
Additionally, menopausal women often have a negative body image, which in turn correlates with low self-esteem (line 55-56).
2.Still some sentences in the text are short and undeveloped e.g. in line 36 … showed that women with a higher severity of vasomotor symptoms experiencing stress, report lower self-esteem [5]. Recommend a “and” between stress, report …
Response:
Thank you for noticing it. Line 36 – we have added ‘and’, now the sentence is:
The results of Hunter and Rental (2007) showed that women with a higher severity of vasomotor symptoms and experiencing stress, report lower self-esteem [5] (line 36).
We do apologize for omitting this word.
We have read the whole texts very carefully and in line 44 – we have also added ‘and’ it the sentence.
3.But otherwise I think the manuscript is better
Response: Thank you very much for your opinion that our manuscript is better now. Undoubtedly it is better thanks to your previous, valuable suggestions and comments. Thank you for your careful and professional review.